# Water Retention in Nature-Based Solutions—Assessment of Potential Economic Effects for Local Social Groups

**Zwoździak Jerzy [1], Szałata Łukasz [2,*], Zwoździak Anna [2], Kwiecińska Kornelia [1] and Byelyayev Maksym [2]**

[1]  Grow Green Project Team, Wroclaw University of Environmental and Life Sciences, 50-375 Wroclaw, Poland; jerzy.zwozdziak@upwr.edu.pl (Z.J.); kornelia.kwiecinska@upwr.edu.pl (K.K.)

[2]  Department of Environment Protection Engineering, Wroclaw University of Science and Technology, 50-377 Wroclaw, Poland; anna.zwozdziak@pwr.edu.pl (Z.A.); 223171@student.pwr.edu.pl (B.M.)

\*  Correspondence: lukasz.szalata@pwr.edu.pl

**Abstract:** The upcoming trends related to climate change are increasing the level of interest of social groups in solutions for the implementation and the realization of activities that will ensure the change of these trends and can reduce the impact on the environment, including the health of the community exposed to these impacts. The implementation of solutions aimed at improving the quality of the environment requires taking into account not only the environmental aspects but also the economic aspect. Taking into account the analysis of solutions changing the current state of climate change, the article focuses on the analysis of the potential economic effect caused by the implementation of nature-based solutions (NBSs) in terms of reducing the operating costs related to water retention for local social groups. The analysis is based on a case study, one of the research projects studying nature-based solutions, created as part of the Grow Green project (H2020) in Wrocław in 2017–2022. The results of the analysis are an observed potential positive change in economic effects, i.e., approximately 85.90% of the operating costs related to water retention have been reduced for local social groups by NBSs.

**Keywords:** nature-based solutions; sustainable development; climate change adaptation; water retention; economic effects

---

## 1. Introduction

"Man can hardly even recognize the devils of his own creation"—Albert Schweitzer. The retention of natural waters and precipitation in urban agglomerations and in highly urbanized areas is an important element in the development of the local economy, especially in the face of changes in the natural environment. In recent years, cities have faced the effects of climate change, i.e., more frequent extreme weather events (hurricanes and heavy rainfall), longer periods of drought and heatwaves, and the extinction of animal and plant species [1–3]. The predominance of impermeable surfaces in the urban landscape affects local environment and results in the exacerbation of problems with rainwater management, low air humidity and the occurrence of urban heat island, especially in summer [4]. Considering the effects of global warming and urbanization, the number of cities affected by surface-water deficits is likely to increase to 27.6% in 2050, where a surface-water deficit means that the available amount of surface water is less than the demand [5]. Therefore, these problems should be tackled as we gain more tools and knowledge to solve them effectively. It is necessary to use comprehensive solutions that would simultaneously answer many problems and would be easily accessible, economically effective, citizen-friendly, and consistent with the principles of sustainable

development [6]. Barking Riverside, London, UK, provides a good example of how some of these issues have been addressed in practice. Barking Riverside is a brownfield development site, where planning consent recognised the importance of the brownfield habitat, as well as its multifunctional ecosystem service values, including stormwater storage, recreation access and biodiversity [7]. Nature-based solutions achieve reliable, permanent and multi-faceted success [8]. They are an excellent solution to problems related to water retention in cities and contribute to increasing the city's adaptation to climate change [9]. The aim of the article is to highlight one of the most difficult and complex issues facing humanity in the 21st century—to build an environment using nature-based solutions (NBSs) in harmony with the inhabitants. The goal is to create a space for dialogue, not to push for utopian solutions that waste natural resources. Empirical evidence suggests that natural water retention measures can be effective in small catchments, but may not have the same effectiveness when up-scaled to larger areas [10]. We must realize that our organized society, i.e., one that has developed a decision-making process of sharing natural resources, has faced the problem of changing climatic conditions. Violent and destructive phenomena, such as excessive pollution emissions to the environment, a lack of respect for nature, land development not taking water retention capacity into account, misunderstood spatial planning and excessive greenhouse gas emissions, have the greatest impact on climate change. Our research shows the economic aspect of combined nature-based solutions creating a rainwater-irrigated green infrastructure.

In official EU documents, nature-based solutions are defined as follows: "Nature-based solutions to societal challenges are solutions that are inspired and supported by nature, which are cost-effective; simultaneously provide environmental, social and economic benefits and help build resilience [11]. Such solutions bring more, and more diverse, nature and natural features and processes into cities, landscapes and seascapes, through locally adapted, resource-efficient and systemic interventions" [12]. Additionally, the EU Thematic Strategy on the Urban Environment recognizes that it is in urban areas that the environmental, economic and social dimensions of the EU Sustainable Development Strategy come together most strongly.

Nature-based solutions in cities include large-scale activities, such as:

- The protection and conservation of existing elements of blue–green infrastructure, as well as the introduction of proper blue–green infrastructure designs into areas dominated by dense development;
- Most activities in the area of the thermo-modernization of buildings;
- The replacement of impervious surfaces with permeable surfaces;
- Recycling and circular economy principles' introduction.

However, they also include local activities, such as:

- The introduction of shelterbelts (vegetative environmental buffers) around industrial plants;
- The introduction of rainwater management systems in courtyards;
- Activities that can be implemented in households, e.g., composting, greywater use and rainwater collection.

Green infrastructure provides numerous benefits for the urban environment—it filters the air, contributes to the reduction of dust, increases air humidity and provides oxygen, retains rainwater, absorbs noise and creates a habitat for many animal species. Green infrastructure is an integrated, complex system that combines individual NBSs. The introduction of green infrastructure to built-up areas allows reducing the urban heat island phenomenon [13]. The accompanying blue (water) infrastructure, i.e., elements related to water, will ensure the retention of rainwater, contribute to an increase in air humidity and increase the biodiversity of urban agglomerations [14,15].

Nature-based solutions are systemic solutions—they bring environmental, social and economic benefits [16]. For nature-based solutions to fit into the urban mosaic, they need to be appealing to citizens, as well as multi-functional [17]. In order to support the implementation of innovative

nature-based solutions in environmental management and land-use planning, valuation becomes essential [18]. Economic effects, as one of the elements important in terms of the implementation of this type of solutions, are the basis for the decision-making process [19]. Some researchers have developed a comprehensive framework for assessing the additional benefits (and costs) of NBSs for components of socio-cultural and socio-economic systems, biodiversity, ecosystems and climate [20]. Economic effects and their evaluation also create local and social budgets [21]. The analysis undertaken in this article focused on the presentation of the economic attractiveness of using NBSs for the society. The results of the research are a real example showing the economic assessment of NBSs in the context of water retention in urban agglomerations. The economic assessment in this scope will make the decision-making process more precise for the implementation of NBSs. Measurable economic effects were estimated in terms of possible losses or financial gains, as well as in terms of reducing operating costs, thus resulting in an economic assessment that justifies the implementation of such solutions.

## 2. Materials and Methods

### 2.1. Site Description

The applied analytical method consisted of determining the potential variability of the retained water amount as a result of the NBS implementation and the related economic effects. Therefore, the research areas and spatial development projects for these areas were selected and analysed in terms of increasing the biologically active surface and increasing the share of rainwater retained by the blue infrastructure.

The research areas were selected as part of the case study analysis, which is the Grow Green project (Horizon 2020) in Wrocław, Poland. The area covered by the project is in the area of the Ołbin district—one of the city's most densely populated districts with a very dense residential development, consisting mainly of century-old tenement houses (mainly municipal housing resource buildings), with internal courtyards, mostly lacking greenery. Besides, due to the predominance of impervious surfaces, the urban heat island in this area is very severe across the whole city; residents also complain about local flooding (especially after heavy rain) and the stagnation of water in the courtyards.

The selected research areas consist of 6 courtyards (generally accessible quarters, mostly surrounded by houses on four sides) and one of the streets (access road)—together, they form a network of demonstrators (project demonstration or research areas). A selected demonstrator project (for the Area No. 1) is presented below (Figure 1). The actual state of the design areas (before the NBS implementation) included impervious surfaces, creating the parking infrastructure. The share of the surface type after the NBS implementation was calculated following the spatial development project; an example is shown in the drawing (Figure 1).

The population in this area is approx. 200 people. The buildings located in the area are buildings of the 19th century, and their technical condition is good. The foundations of the buildings are partially tight. The groundwater level is 1.1 m below the ground level.

According to the spatial development project, following NBSs were implemented in the studied area: biologically active surfaces, i.e., groundcover vegetation, trees and shrubs, wood chips, an animal habitat area, lawn and also elements of blue infrastructure, i.e., a retention basin, rain gardens and ground rainwater tanks with a capacity of 500 $dm^3$ (Figure 2). Rainwater tanks are used to retain rainwater collected from the roof surfaces of buildings, for the purpose of watering the green areas. Rainwater tanks are installed at each gutter, i.e., at 20 gutters. The nature-based solutions also include permeable pavements in some areas, i.e., semi-impervious surfaces—EcoGrid and gravel surfaces.

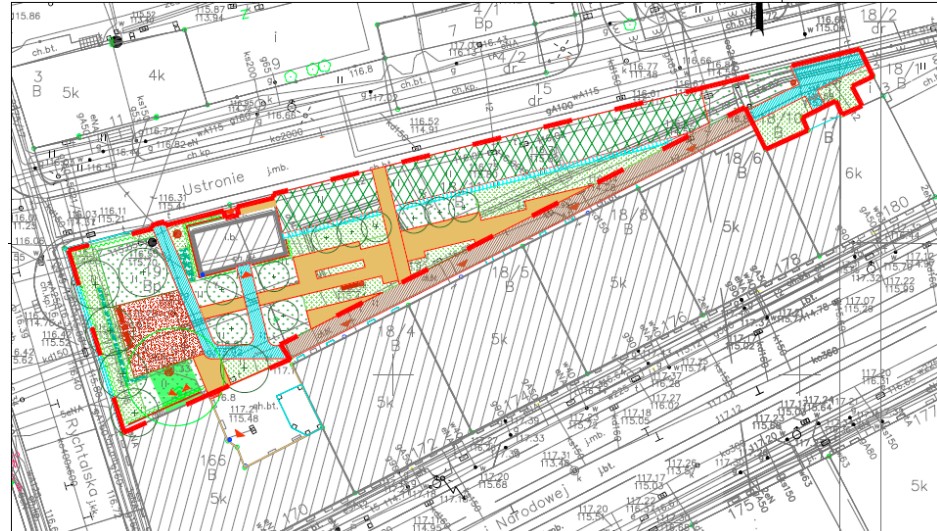

**Figure 1.** Spatial development project for Area No. 1. (Source: Municipality of Wroclaw, author of the project Form.ica Sp. z o.o.).

| Legend | |
|---|---|
| | Existing buildings (roof surface) |
| | An existing sidewalk of cobblestones |
| | Gravel surface |
| | Surface—wood chips |
| | Surface—EcoGrid |
| | Retention basin |
| | Trees and shrubs |
| | Groundcover vegetation |
| | Lawn |
| | Animal habitat area |
| | Rain garden |

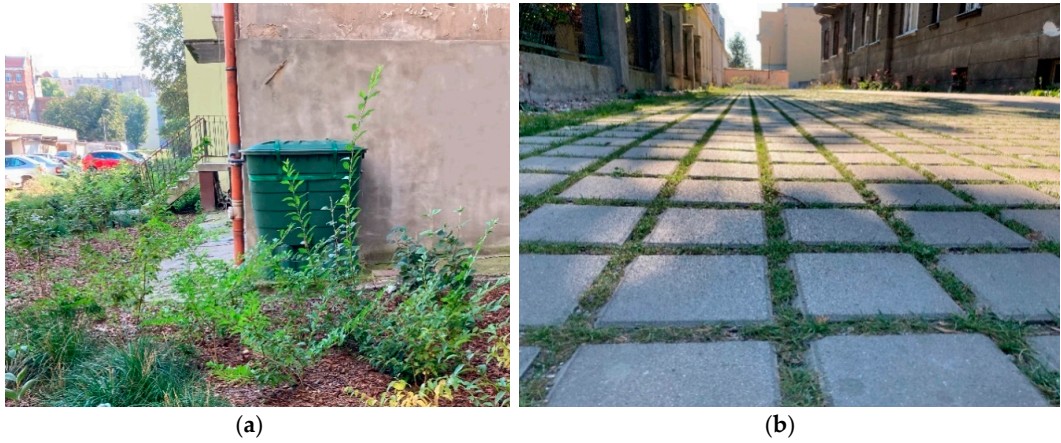

(**a**)                                                (**b**)

**Figure 2.** Selected nature-based solutions (NBSs). (**a**) Rainwater tank—a tank made of polyethylene (PE), characterized by a particular resistance to the effects of UV radiation and resistance to changing

weather conditions. Water intake occurs through the valve. The service life of the tank is approximately 20 years. (**b**) Surface—EcoGrid—a surface made of concrete cubes in an orthogonal system; the gaps are filled with growing medium and grass.

## 2.2. Data Analysis

The potential variability of the retained water amount as a result of the NBSs implementation was estimated based on the change in the amount of rainwater drained into the rainwater sewage system before and after the implementation of the NBSs based on the calculation of the share of biologically active surface and semi-impervious and impervious surfaces. The amount of rainwater drained into the rainwater sewage system was estimated based on hydraulic calculations made for the selected catchment, i.e., the research areas, taking into account the annual amount of precipitation, the surface runoff coefficient, according to the following formula:

$$Q_y = F \cdot \Psi \cdot H_y, \left[ \frac{m^3}{year} \right] \tag{1}$$

where:

$F$ is the catchment area (m$^2$);
$\Psi$ is the surface runoff coefficient (-) (Table 1);
$H_y$ is the annual amount of precipitation, 699 (mm) (source: calculated per year, i.e., from 01/09/2018 to 31/08/2019—compiled by Prof. Ewa Burszta-Adamiak).

**Table 1.** Surface runoff coefficient in accordance with PN-EN 752.

| Type | The Value of the Runoff Coefficient $\Psi$ |
|---|---|
| Impervious surface | $\Psi = 0.90$–$1.00$ |
| Semi-impervious surface | $\Psi = 0.80$–$0.90$ |
| Biologically active surface | $\Psi = 0.00$–$0.10$ |

Impervious surfaces were defined as the surfaces of asphalt roads, concrete roads and buildings roofs, which are characterized by high tightness. Semi-impervious surfaces were EcoGrid and gravel surfaces, characterized by medium tightness. Biologically active surfaces were ground surfaces, lawns, gardens, parks and wood chip surfaces, which are characterized by a high degree of water infiltration.

The economic effects were estimated based on the variability of the operating costs related to the drain rainwater using the sewage system and the government tax for reducing natural surface retention. The cost of rainwater drainage using the sewage system, according to the data of the local sewerage operator, is 2.27 PLN/m$^3$ of drained rainwater (source: fee rate on 01/06/2020, MPWiK—Miejskie Przedsiębiorstwo Wodociągów i Kanalizacji S.A.; in Wrocław – Municipal Water and Sewage Company, Wroclaw), where PLN is the Polish national currency, in accordance with ISO 4217. The cost of government tax is regulated by national regulations, i.e., the provisions of art. 270 sec. 7 and art. 272 paragraph. 8 of the Water Law, where the rules and method for determining the amount of taxes for reducing natural land retention as part of water services are specified [22]. In accordance with paragraph 8 art. 272 of the above-mentioned act: "*The amount of the tax for the reduction of natural land retention caused construction permanently connected with the land on the surface with an area of more than 3500 m$^2$, which reduce this retention by excluding more than 70% of the surface area from the biologically active area in areas not covered by sewage systems open or closed is determined as the product of the unit tax rate, expressed in m$^2$ of the amount of biologically active area lost and the time expressed in years*". The tax and cost of rainwater drainage using the sewage system apply to local community groups living within the property. The amount of this tax, under the Ordinance of the Council of Ministers of December 22, 2017, on the unit rates for water services (Journal of Laws of 2019, item 2452) [23], is as follows: "*Unit*

*tax rates for the reduction of natural land retention caused construction permanently connected with the land on the surface with an area of more than 3500 m², which reduce this retention by excluding more than 70% of the surface area from the biologically active area in areas not covered by sewage systems open or closed sewage systems are":*

(1)　Without water retention devices from impervious surfaces—0.50 PLN/m³/year;
(2)　With infrastructure for the water retention from impervious surfaces with a capacity:

　　(a)　Up to 10% of the annual runoff from impervious areas—0.30 PLN/m³/year;
　　(b)　10% to 30% of the annual runoff from impervious areas—0.15 PLN/m³/year;
　　(c)　Over 30% of the annual runoff from impervious areas—0.05 PLN/m³/year.

The economic effect is represented by a formula showing the degree of the total operating cost reduction related to rainwater retention:

$$E_{ef} = \left(1 - \frac{C_{dr_a} + C_{tax_a}}{C_{dr_b} + C_{tax_b}}\right) \cdot 100\% \tag{2}$$

where the terms are defined as follows:

$C_{dr_a}$—the costs of draining rainwater using the sewage system after implementing the NBS, $\left[\frac{EUR}{year}\right]$;

$C_{tax_a}$—government tax after implementing the NBS, $\left[\frac{EUR}{year}\right]$;

$C_{dr_b}$—the costs of draining rainwater using the sewage system before implementing the NBS, $\left[\frac{EUR}{year}\right]$;

$C_{tax_b}$—the government tax before implementing the NBS, $\left[\frac{EUR}{year}\right]$.

These operating costs are regulated solely by the local social group. The above formula for estimating the economic effect was used to show a measurable change in the economic effect caused by rainwater retention, as determined for the local social groups.

## 3. Results

According to the selected analytical method, using the Formula (1), the amount of rainwater drained from the research areas was calculated, taking into account the surface runoff coefficient and operating costs (Table 2).

Due to the implemented blue infrastructure solutions, i.e., tanks for rainwater collected from the roofs of buildings, roof surfaces were selected as a separate surface to calculate the potential benefits related to the reduction of rainwater drained using the sewage system. It was assumed that the implemented rainwater tanks would accumulate all the atmospheric precipitation collected from the roofs of buildings. According to the selected analytical method, the potential changes in economic effects related to the implementation of the NBSs for social groups were calculated using the Formula (1) (Table 3).

**Table 2.** The cost of draining rainwater and the government tax depending on the amount of drained water.

| Research areas | Impervious surface[g] (m²) | Roof surface[g] (m²) | Semi-impervious surface[h] (m²) | Biologically active surface[j] (m²) | The amount of drained rainwater using the sewage system [m³/year] | Costs of draining rainwater, EUR[k]/year | Government tax, EUR[k]/year |
|---|---|---|---|---|---|---|---|
| **The actual state—before implementing the NBS** | | | | | | | |
| Street | 15,348.00 | 0.00 | 0.00 | 0.00 | 10,728.25 | 5411.81 | 1192.03 [1] |
| 1 | 1506.80 | 1680.00 | 0.00 | 0.00 | 2227.57 | 1123.69 | 247.51 [1] |
| 2 | 1499.60 | 2368.40 | 0.00 | 0.00 | 2703.73 | 1363.88 | 300.41 [1] |
| 3 | 6718.80 | 5643.00 | 0.00 | 0.00 | 8640.90 | 4358.85 | 960.10 [1] |
| 4 | 6691.60 | 9067.20 | 0.00 | 0.00 | 11,015.40 | 5556.66 | 1223.93 [1] |
| 5 | 88.80 | 0.00 | 0.00 | 1851.20 | 62.07 | 31.31 | 6.90 [1] |
| 6 | 3170.00 | 2016.80 | 0.00 | 0.00 | 3625.57 | 1828.90 | 402.84 [1] |
| 7 | 3169.20 | 2668.40 | 0.00 | 0.00 | 4080.48 | 2058.38 | 453.39 [1] |
| **Projected state—after implementing NBS** | | | | | | | |
| Street | 14,183.44 | 0.00 | 607.00 | 554.56 | 9914.22 | 5001.18 | 1101.58 [1] |
| 1 | 410.80 | 1680.00 | 169.60 | 926.40 | 381.99 | 192.69 | 4.24 [2c] |
| 2 | 176.00 | 2368.40 | 83.60 | 1240.00 | 169.77 | 85.64 | 1.89 [2c] |
| 3 | 2630.80 | 5643.00 | 476.80 | 3612.00 | 2105.56 | 1062.14 | 23.40 [2c] |
| 4 | 1114.40 | 9067.20 | 910.00 | 4667.20 | 1287.84 | 649.64 | 14.31 [2c] |
| 5 | 88.80 | 0.00 | 0.00 | 1851.20 | 62.07 | 31.31 | 0.69 [2c] |
| 6 | 1044.80 | 2016.80 | 83.20 | 2042.00 | 776.84 | 391.87 | 8.63 [2c] |
| 7 | 1003.60 | 2668.40 | 214.40 | 1951.20 | 821.41 | 414.36 | 9.13 [2c] |

[g] the assumed surface runoff coefficient is 1.00|[h] the assumed surface runoff coefficient is 0.80|[j] the assumed surface runoff coefficient is 0.00|[k] the adopted EUR/PLN exchange rate is 4.50|[1,2,c] tax category.

**Table 3.** Economic effects related to the implementation of the NBS.

| Research Areas | Amount of Total Reduced Operational Cost, EUR/Year | Economic Effect, % | Amount of Total Reduced Operational Cost per Unit Area, EUR/m$^2$/Year |
|---|---|---|---|
| Street | 501.08 | 7.59 | 0.03 |
| 1 | 1174.26 | 85.64 | 0.37 |
| 2 | 1576.77 | 94.74 | 0.41 |
| 3 | 4233.42 | 79.59 | 0.34 |
| 4 | 6116.64 | 90.21 | 0.39 |
| 5 | 0.00 | 0.00 | 0.00 |
| 6 | 1831.24 | 82.05 | 0.35 |
| 7 | 2088.28 | 83.14 | 0.36 |

The "Street" research area is an access road that connects the streets, and the impervious surface is unchangeable in terms of changing its functions. Therefore, the economic effect related to the cost of rainwater drainage for the "Street" area is 7.59%, which is a small value compared to other areas (Figure 3). Additionally, Research Area No. 5, in which the economic effect was 0.00%, is an area in which the configuration of the NBSs consists solely of increasing the biodiversity of the existing green area, which results in no economic effect related to the cost of rainwater drainage.

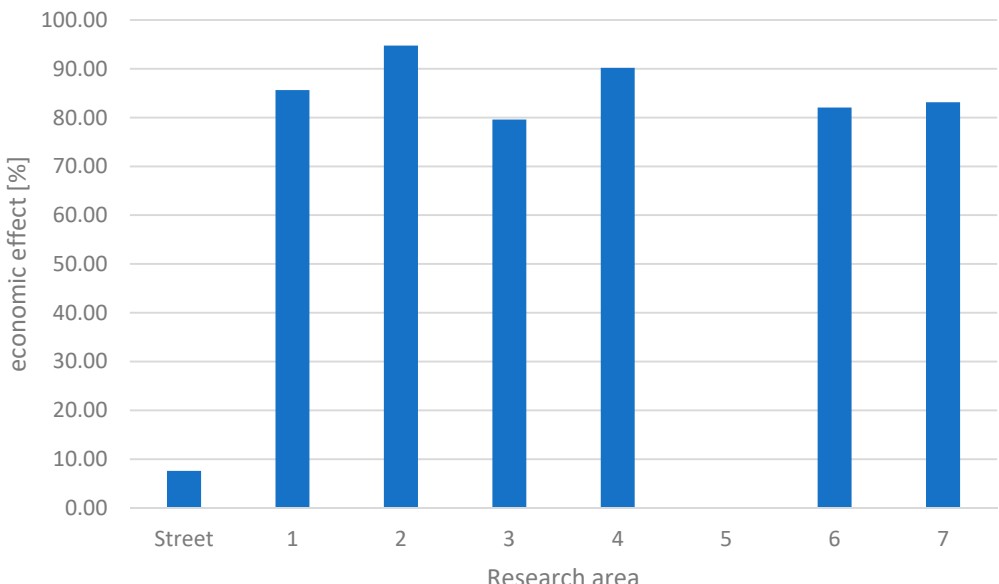

**Figure 3.** Summary of the economic effect for all research areas.

The remaining design areas are mostly a uniform conceptual system, i.e., all these areas are municipal housing resource buildings in multi-family buildings. Classifying a data set, i.e., research areas, according to their common features, i.e., the layout and functions of the areas, a large group was designated with common features, which includes Area No. 1, 2, 3, 4, 6 and 7. For this data set, the arithmetic mean of the total reduced operational cost per unit area was determined, which is 0.37, which in terms of the economic effect is 85.90%, with the standard deviation of the tested data set being 0.02.

## 4. Conclusions and Discussion

During the implementation of the project, the main problem that constantly arose was the lack of social acceptance for the changes that were to take place. Social acceptance appeared after conducting a series of meetings with the local community, during which the benefits of green infrastructure were

demonstrated, with a particular emphasis on its impact on survival in the conditions of a changing climate, i.e., climate change adaptation and cultural and civilization development.

A disadvantage in the project's implementation may be the lack of the sociological preparation of the local community. In the future, deterministic modelling should be introduced before the implementation of this type of project, which will allow determining the time and period acceptable for the local community for a given project.

The twentieth century community has distanced itself from nature and the environment more than ever before. The presented project is an opportunity to change this condition. With an apocalyptic vision of a world ravaged by climate change, nature-based solutions, including complex water retention solutions, become the basis for changing this vision and developing tools to build climate change adaptation.

The implementation of NBSs in the form of green and blue infrastructure results in positive economic effects related to the reduction of operating costs caused by rainwater retention for local social groups. The operating costs related to the rainwater drainage using the sewage system and the government tax for reducing natural surface retention incurred by the local social group living in the areas have been reduced, on average, for all areas by 0.37 EUR/m$^2$/year. The observed economic effects related to the rainwater retention, amounting to an average of 85.90%, indicate the sense of implementing NBSs, which helps to clarify the decision-making process for social groups of large urban agglomerations. Besides, the obtained results may be used in decision-making models based on the principles of sustainable development and also taking into account environmental aspects related to the implementation of NBSs. The analysed spatial conditions, i.e., the architectural infrastructure for the studied areas, led to the acceptance of the results in question on a European scale.

The conducted analysis and results can create a benchmark for the community in terms of planning and building local budgets, as well as for estimating the total economic profitability of NBSs. The potential possible changes in the economic effects observed in this analysis were based on real projects in line with the design stage. The implementation of NBSs following the analysed projects may lead to the designated positive effects, while the actual effects determined based on environmental research will be realized in further research progress.

In general, the presented activities undoubtedly provide us with benefits resulting from the use of innovative solutions in the construction of decision trees and in making decisions and guide us in which direction to lead the development of adaptation to climate change.

**Author Contributions:** Conceptualization, Z.J. and S.Ł.; methodology, B.M. and S.Ł.; software, B.M.; validation, Z.A., K.K. and Z.J.; formal analysis, S.Ł.; investigation, K.K.; resources, J.Z. and K.K.; data curation, Z.A.; writing—original draft preparation, Ł.S., Z.A. and B.M.; writing—review and editing, J.Z.; visualization, B.M. and S.Ł.; supervision, J.Z.; project administration, K.K.; funding acquisition, Z.A. All authors have read and agreed to the published version of the manuscript.

**Funding:** This research received no external funding.

**Acknowledgments:** 

**Conflicts of Interest:** The authors declare no conflict of interest.

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
