# Peer review of "Water Retention in Nature-Based Solutions—Assessment of Potential Economic Effects for Local Social Groups"

_water, doi:10.3390/w12123347_

Round 1

Reviewer 1 Report

The authors introduced a nature-based solution of urban water retention in European and perform the cost-efficient analysis based on the tax schemes. This manuscript is straight forward and clear. Beside the advantage, there are some suggestions might help the readers to understand the benefit of NBS.

  • Modify the abstract and let numbers talk.
  • Modify the introduction section and list the similar cases worldwide. Distinguish the differences between previous study and current project. Highlight the necessary of this study.
  • Improve Figure 1 and make it readable to the ordinary reader.
  • Explain PLN before using abbreviation. Other comments can be found in the attachment.

Author Response

Dear reviewer

All comments were included in the redacted file. The file contains comments from other reviewers.

Regarding the remark on "relation between street and 1-7", research areas 1-7 and Street - these are project areas implemented within the project.

Research areas are methodologically selected in the city of Wrocław, in accordance with the assumed project result. Generally, there is no relation of areas in terms of water retention effects. The concept of combining design areas stems from an air impact that is not relevant to this article.

Best regards,

Authors

Reviewer 2 Report

The paper “Water retention in Nature-Based solutions – assessment of potential economic effects for local social groups” is an empirical study aiming at assessing the potential economic effects of the implementation of nature-based solutions in 7 different areas of Wroclaw. The use of these nature-based solutions responds to a need for sustainable transition regarding current development and future global change, e.g., population growth, climate change, land changes.

The paper is well-written, fits the scope of Water and provide clear results. Nonetheless, revisions are necessary before it is suitable for publication, especially regarding the conceptual part and the methodology. Please find below my remarks.

General comments

-There is a need to significantly develop the literature review to better present what are the current challenges, the potential for nature -based solutions, what is known and ignored and what is your contribution. Especially, please elaborate more on the future challenge related to climate change in cities and the potential for Nature based solution. For instance, there are many studies assessing future urban water deficit (e.g. McDonald et al. 2013, Florke et al. 2018), how water-related disasters associate with development patterns (e.g. Bolognesi, 2015), etc. This literature emphasizes the need for a transition and Nature Based Solutions could be part of it (here is your contribution). Besides, we need to know more about Nature-based solutions, what expected economic benefits (e.g. Liquete et al. 2016, Raymond et al. 2017); lessons learned (e.g. Frantzeskaki, 2019); cases, the main grey area in the knowledge, etc.

-The introduction does not provide a straightforward research question. What is your contribution to the current knowledge? What is the goal of the paper? What hypotheses do you test, and why is it important?

-Explain why your case is relevant in general, not solely in regard to the Wroclaw urban area.

-If I understood well the paper, you measure the expected/likely economic effects based on the parameter assumptions made in table 2. You must be crystal clear about what you do. Until this table, I thought you were calculating the actual economic effects. And after the table, this confusion is still present. For instance, table 3 should be entitled: “Expected economic effects related…”

-Results present the expected economic effect in relative terms. I recommend a discussion of the absolute value of the economize costs as well. Indeed, if we want to consider the incentivization of adopting such NBS, we need to know the opportunity costs. How long it takes to build a green infrastructure? What are the main nuisances?

I would discuss further the role of tax in equation 2. To some extent, it is likely that the taxes are not a cost at 100%. Indeed, the State could reallocate this sum, especially if a public authority makes the investment.

Minor comments

l.63: what is the difference between green infrastructure and nature-based solution?

Equation 1: is this equation a common use, is there alternatives? Please discuss the measurement. Clarify what is Qy

Table 1: where did you get these coefficients? Are they derived from direct measurement you made or from existing studies?

I recommend to draw graphs showing the evolution of Eef and Qy. They are your central measurement but we know little about them.

l.233: Authors should not sign their paper.

The authors often refer to an H2020 project that they seemingly carried out. Please, do not and focus on your case instead. It is not necessary to know that your case is part of a project.

References that could help.

McDonald, Robert I., Katherine Weber, Julie Padowski, Martina Flörke, Christof Schneider, Pamela A. Green, Thomas Gleeson, et al. 2014. ‘Water on an Urban Planet: Urbanization and the Reach of Urban Water Infrastructure’. Global Environmental Change 27: 96–105. https://doi.org/10.1016/j.gloenvcha.2014.04.022.

Flörke, Martina, Christof Schneider, and Robert I. McDonald. 2018. ‘Water Competition between Cities and Agriculture Driven by Climate Change and Urban Growth’. Nature Sustainability 1 (1): 51–58. https://doi.org/10.1038/s41893-017-0006-8.

Bolognesi, Thomas. 2015. ‘The Water Vulnerability of Metro and Megacities: An Investigation of Structural Determinants’. Natural Resources Forum 39 (2): 123–133. https://doi.org/10.1111/1477-8947.12056.

Raymond, Christopher M., et al. "A framework for assessing and implementing the co-benefits of nature-based solutions in urban areas." Environmental Science & Policy 77 (2017): 15-24. https://doi.org/10.1016/j.envsci.2017.07.008

Liquete, Camino, et al. "Integrated valuation of a nature-based solution for water pollution control. Highlighting hidden benefits." Ecosystem Services 22 (2016): 392-401. https://doi.org/10.1016/j.ecoser.2016.09.011

Frantzeskaki, N. (2019). Seven lessons for planning nature-based solutions in cities. Environmental science & policy, 93, 101-111. https://doi.org/10.1016/j.envsci.2018.12.033

Author Response

Dear reviewer

All comments have been taken into account with the exception of optional changes. The changes are in the redacted file. The file contains comments from other reviewers.

Regarding "..measure the expected / likely economic effects ..", it should be noted that the article was prepared for the actual state, ie after implementation.

The data presented in the article show the previous and present status. As an analytical method, the overall execution of the project was considered in accordance with the design assumption, which is indeed the case.

Unfortunately, at the moment we do not have data on investment costs related to the construction of these solutions, because another local government unit is responsible for the execution. The absolute costs should be analyzed on the basis of invoices received from residents (such data is difficult to obtain).

Taxes discussed in equation 2 are paid by residents, especially in Poland. The Polish state in no way compensates for these expenses by the inhabitants. Therefore, financial expenses incurred by residents related to taxes, included in the costs.

NBS and green infrastructure differs in the scale of implementation, ie NBS is a solution used in the area of one facility, and the green infrastructure is an integrated complex system that combines individual NBS solutions.

Formula 1 presented in the article, i.e. the general formula for calculating rainwater runoff.

The coefficients in Table 1 are surface runoff coefficients, which shows the ratio of the amount of rainwater runoff from the catchment area to the amount of rainfall in this catchment area. The value of the runoff coefficient depends mainly on the type of development (sealing) of the catchment area in accordance with PN-EN 752.

Best regards,

Authors

Reviewer 3 Report

The issue of the possibility of retention in urbanized areas in the context of climate change is very important. This paper proposes focuses on the analysis of the potential economic effect caused by the implementation of Nature-Based Solution (NBS) in terms of reducing operating costs related to water retention for local social groups. The proposal does fit in Water journal but the article is very laconic and lacks a strict methodological foundation. It is mainly a contribution to a simplified analysis relating to the local case study.. Neither the introduction nor the results and discussion links with the wide-ranging problem of adaptation to climate change.
The paper lacks information and explanations of certain concepts (e.g. explaining the meaning of blue-green infrastructure designs, what is MPWiK).
With regard to the content of the paper, the question arises whether we are able to build such tanks everywhere in the city to collect all rainfall from the roofs. What about heavy rains?
The work presents only theoretical considerations on the impact of introducing NBS in urban areas. An example of regional (Polish) regulations indicating the profitability of such activities is presented.
What the other cost components look like: e.g. investment, depreciation, operating costs, cost of water use (cleaning, pumping)?
The conclusions and Discussion are disappointing and vague. It has to do with the fact that a clear research question or objective is lacking. The latter shows from the Abstract where there is no indication of the research question/objective.
The work lacks Discussions - references to the results of other research by scientists. Indication of places where such a solution has been implemented. What is the actual percentage savings? What problems have arisen during operation, etc.? This is the basic problem of this work, which does not meet such standards.
I would want to invite the authors to rewrite both the Abstract and the Conclusions and Discussion section. The abstract I suggest should follow the - SCARN format: Situation – Complication – Approach – Results – Next. Making more concrete conclusions and recommendations, preferably based on some calculations would make this manuscript so much more interesting and convincing!

Author Response

Dear reviewer

All comments have been taken into account with the exception of optional changes. The changes are in the redacted file. The file contains comments from other reviewers.

Water tanks and infrastructure ensure the possibility of receiving heavy rains. The article in question was developed for the actual state, ie after implementation.

The data presented in the article show the previous and present status. As an analytical method, the overall execution of the project was considered in accordance with the design assumption, which is indeed the case.

Unfortunately, at the moment we do not have data on investment costs related to the construction of these solutions, because another local government unit is responsible for the execution. The absolute costs should be analyzed on the basis of invoices received from residents (such data is difficult to obtain).

Best regards,

Authors

Round 2

Reviewer 1 Report

Thank you for responding to most the comments. Please take care of some minor issues before publishing. 

1. check line 124, "or ie" ?

2. Please add real color map with coordinator for Area #1.

3. check line 140 "ie. 20" ?

4. Please convert PLN to Euro since most text is using Euro. 

Reviewer 2 Report

The authors have addressed most of the comments. The paper is clearer.

- The sentence on l.38-39 must be rephrased or quated properly as it comes from the Flörke et al.'s paper (in reference 7).

- A few typos remain, e.g. p. 55, 90.

Best

Reviewer 3 Report

The authors revised their manuscript. Most of the comments and suggestions were taken into account. This should be appreciated. The article can be possibly recommended for publication, after taking into account comments and especially a better reference to other work and results obtained by other researchers (especially in the last chapter where the discussion is). This is still missing.